



# Influence of extreme events modeled by Lévy flight on global thermohaline circulation stability

Daniel Tesfay[1], Larissa Serdukova[2], Yayun Zheng[1], Pingyuan Wei[1], Jinqiao Duan[3], and Jürgen Kurths[4]

[1]School of Mathematics and Statistics & Wuhan Center for Mathematical Sciences, Huazhong University of Science and Technology, Wuhan 430074, China
[2]School of Mathematics, Georgia Institute of Technology, Atlanta, USA
[3]Department of Applied Mathematics, Illinois Institute of Technology, 312-567-5335, Chicago 60616, USA
[4]Research Domain on Transdisciplinary Concepts and Methods, Potsdam Institute for Climate Impact Research, PO Box 60 12 03, 14412 Potsdam, Germany

**Correspondence:** Larissa Serdukova (larissa.serdukova@math.gatech.edu)

**Abstract.**

How will extreme events due to human activities and climate change affect the oceanic thermohaline circulation is a key concern in climate predictions. The stability of the thermohaline circulation with respect to extreme events, such as fresh-water oscillations, greenhouse gas accumulations and collapse of the Atlantic meridional overturning circulation, is examined

using a conceptual stochastic Stommel two-compartment model. The extreme fluctuations are modeled by symmetric $\alpha$-stable Lévy motions whose pathways are cádlág functions with at most a countable number of jumps. The mean first passage time, escape probability and stochastic basin of attraction are used to perform the stability analysis of *on (off)* equilibrium states. Our results argue that for model with weak fresh-water forcing strength, the greatest threat to the stability of the *on*-state represents noise with low jumps and higher frequency that can be seen as civilization-induced greenhouse gas accumulation.

On the other hand, the *off*-state stability is more vulnerable to the agitations with moderate jumps and frequencies which can be interpreted as wind-driven circulations towards higher latitudes. Under the repercussion of stochastic noise, *on* to *off* transitions are more expected in the model if the fresh-water influx is strong. Moreover, transitions from one metastable state to another are equiprobable when the fresh-water input induces a symmetric potential well.

## 1 Introduction

Natural and civilization catalyzed fluctuations in climate have significant impact on the ocean and ocean circulation pattern variations greatly affect climate (Chapman and Shackleton, 1999; Clark et. al., 2002). The global thermohaline circulation (THC), known as great ocean conveyor as well, consists of ocean tides and drifts and loops that have extensive ramifications on climate. THC is basically an outcome of the interplay of fresh-water with thermal energy along with the ocean-atmosphere interface and inside the ocean competition of temperature and salinity (Rahmstorf, 2003). This enormous oceanic process has

a significant contribution in maintaining the equilibrium of Earth's energy framework by redistributing thermal energy of the order $1.2 \times 10^{15}$W northwards in the Atlantic ocean. A large proportion of the meridional overturning circulation (MOC) is usu-





ally categorized as THC because MOC takes the lion's share in this heat penetration to the north pole (Ganachaud and Wunsch, 2001; Trenberth and Solomon, 1994). The warm and saltier surface water on interannual and even longer time-scales gets freshened and loses heat to the cold atmosphere. Subsequently, the water descends slightly to the bottom of the Atlantic since it gets

denser than the underneath water. The cooled water eventually returns southward as deep current and the warm temperature around the equatorial belt opts for upwelling. This thermal forcing is deterred by the surplus of rainfall and water discharge over evaporation at the low latitudes which instigates a fresh-water flux opposing the descend in and around the north pole. The global THC is a combination of the floating of deepwater currents around the equator and in the southern oceans, the horizontal currents, and the descending and forming of deepwater in high latitudes. Climate reconstructions indicate that con-

veyor belt has indispensable contribution in the climate system transitions from cold to warm or from warm to cold climate states (Rahmstorf, 1995; Bond et.al., 1997; Grootes and Stuiver, 1997). Studying the stability of this oceanic conveyor belt by analyzing the influence of internal and external agitations on its dynamical behavior is increasingly pressing presently.

To study how the THC transports properties latitudinally, a conceptual deterministic two-compartment model was forwarded by Stommel (Stommel, 1961). Shreds of evidence from sea observation and model simulation show that the strength of the

thermohaline flow is sensitive to the surface fresh-water flux fluctuation (Jackson and Wood, 2017; Caesar et.al., 2018). Competition between Thermal versus saline forcing competition can lead to a multiple equilibria regime if the relaxation time-scales for the temperature and the salinity are distinct. The thermohaline flow system is bistable, one with strong circulation (analogous with the present set up), and the second state with a very weak flow, when the salinity difference is forced by a prescribed fresh-water flux (Marotzke, 2000). The multistability of THC is also verified by results obtained from different numerical

models (Broecker, 1987). The deterministic compartment model has been further extended to include noisy thermal and saline forcing oscillations (Huang et. al., 1992; Rahmstorf, 1996; Djikstra, 2005).

Forcing the global general ocean circulation model with some particularly large stochastic fresh-water fluctuations is found to trigger pulsation of transport from one stable configuration to the other (Mikolajewicz and Maier-Reimer, 1990). The Gaussian noise perturbed THC has been under extensive study. For instance, it was shown in (Vélez-Belchí et. al., 2001) that an

increment of 5% of the fresh-water forcing in the THC could stimulate transitions between a high and low salinity difference metastable states. In a time-dependent compartment model for THC with Brownian motion and moderate noise intensity, hysteresis does not adiabatically follow stationary distribution (Bergund and Gentz, 2002). Meanwhile, the noise forcing climate comprises of a non-Gaussian $\alpha$-stable Lévy noise component (Fuhrer et. al., 1993; Ditlevsen, 1999). The occurrence more than a dozen of additional Dansgaard-Oeschger (D-O) events that took place during the last glacial period could not have been

reproduced by using the Brownian process. Other extreme events such as stock market crashes in the financial industry, strokes and seizures, and earthquakes are not continuous perturbation processes. The jumps in those events could better be modeled by Lévy flights (Kuhwald and Pavlyukevich, 2016).

Paleoclimatic data indicate the coincidence of transitions from strong to weak or from weak to strong thermohaline circulation states with the occurrence of extreme climatic variations (Vélez-Belchí et. al., 2001). We analyze the stability of metastable

states of the THC model against extreme events modeled by stable Lévy motion by calculating three quantities, namely, mean first passage (exit) time, first passage (escape) probability and stochastic basin of attraction that carry the dynamical information





of the model. In the present form of the THC, analyzing the intensity and mechanism of the forcing schemes that could trigger such transitions and studying the stability of strong THC and weak THC equilibrium states is of fundamental importance.

Particularly, we will study the effect of extreme events on the scalar stochastic THC model

$$dY_t = -V'(Y_t)dt + dL_t^\alpha \tag{1}$$

by measuring the stability of equilibrium states of the salinity difference process $Y_t$ for various values of (nondimensional) fresh-water forcing and non-Gaussianity parameter $\alpha$. In Eq. (1), $V$ is a potential function (details are given in Section 2).

The paper is structured as follows. In Section 2, we discuss the simplified conceptual stochastic Stommel two-compartment model for thermohaline circulation. A brief introduction of the stability analysis measures is provided in Section 3. Stability analysis of the stochastic thermohaline circulation system is given and results obtained are presented in Section 4. Our findings are summarized in Section 5.

## 2   Oceanic thermohaline circulation model

Global thermohaline circulation is an oceanographic phenomena that refers to the movement of ocean waters across both hemispheres and is responsible for the heat transfer and redistribution, acting as a regulator of the global climate. The schematic functioning of THC is shown in Fig. 1. The main engine of this circulation is the difference in density between ocean currents $\Delta\rho$, which is determined by the salinity $S_e, S_p$ and the temperature $T_e, T_p$ of the water and can be represented by

$$\Delta\rho = \beta_S(S_e - S_p) - \beta_T(T_e - T_p), \tag{2}$$

where $\beta_T = 0.17 \times 10^{-3}\,{}^o\,C^{-1}$ is the thermal expansion coefficient and $\beta_S = 0.75 \times 10^{-3}\,\text{psu}^{-1}$ is the haline contraction coefficients, respectively. The surface ocean waters in the subtropical regions due to intense evaporation $F_s/2$ have high salinity $S_e$, however the high water temperature $T_e$ maintains the low density and prevent surface waters sinking.

In high latitude areas, the formation of dense water is mainly associated with lower temperatures $T_p$ and increased salinity $S_p$ due to the formation of ice. Thus, in the polar regions, the increase in surface water density causes it to sink and displace deep water. In this way, the origin of the thermohaline circulation is a vertical flow of surface water $\frac{1}{2}Q(\Delta\rho)$, diving to an intermediate depth or close to the bottom, depending on the density of that water. The systems of superficial and deep circulation of the oceans are interconnected. The continuation is a horizontal flow: the recently sunk waters repel in the horizontal direction the deep waters that occupied this place. In this way, the cold, dense waters sink and slowly flow towards the equator. Thermal energy and salinity balances can be defined by the system of differential equations (Cessi, 1994) (the dots represent derivative


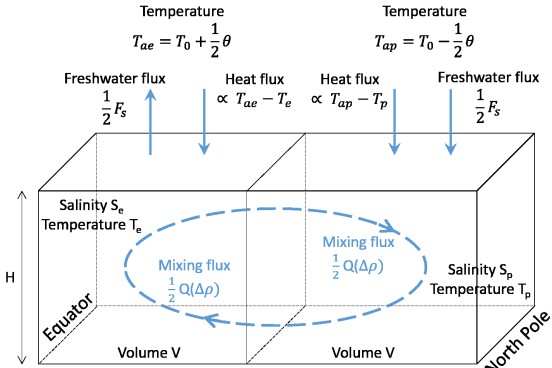

**Figure 1.** The two-compartment model of Stommel adapted from (Cessi, 1994). Each compartment represents the waters of the equatorial and polar oceans with the same volumes $V = 300 \times 4.5 \times 8{,}250$ km$^3$ and temperature $T_e/T_p$ and salinity $S_e/S_p$ characteristic of each of them. The other system parameters are the mean ocean depth $H = 4500$ m, the exchange mass function $Q$, the density gradient $\Delta\rho$, the freshwater flux $F_s$, the equatorial atmospheric temperature $T_{ae}$, the polar atmospheric temperature $T_{ap}$, the reference temperature $T_0 = 5^o$ C, the reference salinity $S_0 = 35$ psu, the meridional temperature difference $\theta = 25$ K and the temperature relaxation time scale $t_r = 25$ days.

with respect to time):

$$\dot{T}_e = -t_r^{-1}(T_e - (T_0 + \frac{\theta}{2})) - \frac{1}{2}Q(\Delta\rho)(T_e - T_p),$$

$$\dot{T}_p = -t_r^{-1}(T_p - (T_0 - \frac{\theta}{2})) - \frac{1}{2}Q(\Delta\rho)(T_p - T_e),$$

$$\dot{S}_e = \frac{F_S}{2H}S_0 - \frac{1}{2}Q(\Delta\rho)(S_e - S_p),$$

$$\dot{S}_p = -\frac{F_S}{2H}S_0 - \frac{1}{2}Q(\Delta\rho)(S_p - S_e), \tag{3}$$

where $Q(\Delta\rho) = t_d^{-1} + V^{-1}q(\Delta\rho)^2$ is the exchange mass function with diffusive time scale $t_d = 180$ years between the two compartments and transport coefficient $q = 1.92 \times 10^{12}$ m$^3$s$^{-1}$. The other parameters of the system are defined in the caption

of Fig. 1.

The time evolution of temperature $\Delta T \equiv T_e - T_p$ and salinity difference $\Delta S \equiv S_e - S_p$ between the compartments are obtained by subtracting the conservation equations (3), respectively.

$$\frac{d\Delta T}{dt} = -t_r^{-1}(\Delta T - \theta) - Q(\Delta\rho)\Delta T,$$

$$\frac{d\Delta S}{dt} = \frac{F_S}{H}S_0 - Q(\Delta\rho)\Delta S. \tag{4}$$




The original system (4) with the substitutions    $x \equiv \frac{\Delta T}{\theta}$,    $y \equiv \frac{\Delta S \beta_s}{\theta \beta_T}$,    $\tau \equiv \frac{t}{t_d}$ is reduced to the dimensionless system of evolution equations (Cessi, 1994; Tesfay et. al., 2020)

$$dx = (-\beta(x-1) - x[1 + \mu^2(x-y)^2])d\tau,$$
$$dy = (F - y[1 + \mu^2(x-y)^2])d\tau, \tag{5}$$

where $\beta$ the temperature restoration tensility, $\mu^2$ the buoyancy-driven convection strength and $F$ dimensionless fresh-water
forcing are defined as

$$\beta = \frac{t_d}{t_r}, \qquad \mu^2 = \frac{q t_d \beta_T \theta^2}{q t_d \beta_T \theta^2}, \qquad F = \frac{\beta_S S_0 t_d}{\beta_T \theta H} F_S. \tag{6}$$

The dynamical system (5) can be further simplified, since the diffusion time scale $t_d$ is much larger than the temperature-restoring time scale $t_r$. Thus, taking the approximation $x = 1 + \mathcal{O}(\beta^{-1})$, we get the first order differential equation in $y(t)$

$$dy = (F - y[1 + \mu^2(1-y)^2])dt. \tag{7}$$

where $\tau$ is replaced by the usual notation of time $t$ for convenience. Considering that $F$ is constant, Eq. (7) can be written as $dy = -V'(y)dt$ with the potential function

$$V(y) = \mu^2 \left( \frac{y^4}{4} - \frac{2}{3}y^3 + \frac{y^2}{2} \right) + \frac{y^2}{2} - Fy. \tag{8}$$

The variation in the fresh-water forcing strength $F$, Fig 2 (b), suddenly changes the qualitative behaviour of the THC system
giving rise to the two bifurcation points $F = 0.9556$ and $F = 1.2963$. For $F < 0.9556$ the system has a single stable equilibrium point, called *on*-state, Fig 2 (a). In this state $y$ is small and this matches with relatively large equator to north pole heat transport. For $0.9556 < F < 1.2963$, Fig 2 (d), (e) and (f) in addition to the stable *on*-state, two more equilibria emerge: one of which is stable, called *off*-state and the other is unstable one. In the *off*-state $y$ is large and corresponds to weak (or even reversed) circulation. Only stable *off*-state takes place in the THC systems with $F > 1.2963$, Fig 2 (c). With global warming, by the end
of the twenty-first century, the mean surface air temperature due to harmful human activities leading to the accumulation of greenhouse gases in the atmosphere will increase by $2 - 6^o$ C (Chapman and Shackleton, 1999; Ganachaud and Wunsch, 2001; Rahmstorf, 2003). This large-scale warming of the climate can cause an increase in frequency or/and intensity of extreme events on the time-scale of decades, such as high-latitude precipitations, the Greenland ice sheet deglaciation, the freshwater outflows to the oceans that certainly affect the dynamic aspects of thermohaline circulation (Clark et. al., 2002; Jackson and Wood, 2017;
Gregory et. al., 2017). An increase in the mass of freshwater in the global ocean reduces its density and thereby complicates its deep immersion, which can slow down THC and even "switch it off" if the parameters cross the tipping threshold (Stommel, 1961). Such extreme and fast events cannot be modeled by deterministic models, since they do not take into account the uncertainty, unpredictability and the likelihood of their occurrence. The Brownian motion predicts the behavior of such random fluctuation with very low accuracy, since it has continuous sample paths and normally distributed increments. It was proved
(Ditlevsen, 1999; Marotzke, 2000) that the Lévy process, characterized by heavy-tailed distribution and discontinuous cadlag

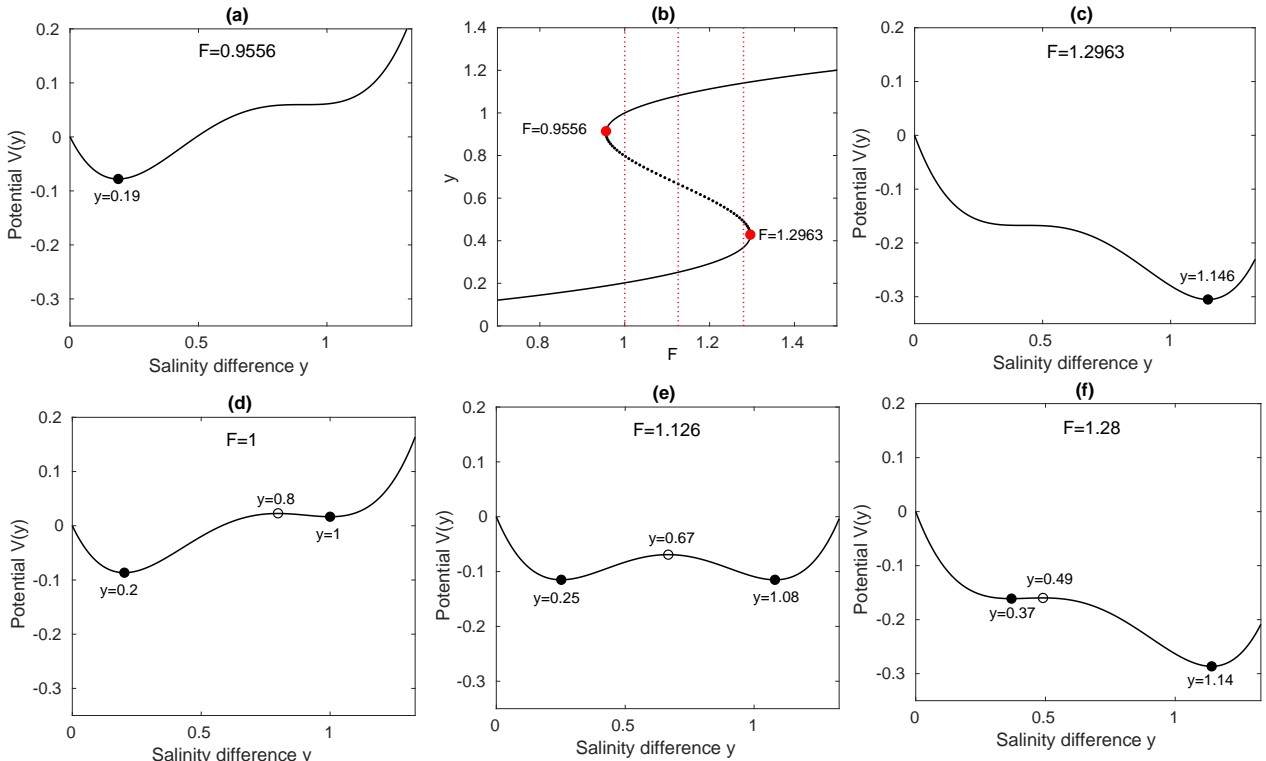

**Figure 2.** The Stommel two-compartment model for the thermohaline circulation may have multiple equilibria. (b) The bifurcation diagram for the salinity difference $y$, as a function of fresh-water forcing strength $F$. The potential function $V(y)$ for forcing strength (a) $F = 0.9556$, (c) $F = 1.2963$, (d) $F = 1$, (e) $F = 1.126$ and (f) $F = 1.28$.

paths, with a certain precision simulates and predicts these rare events. Consider now that the fresh-water flux can be written as sum of a stochastic component $dL_t^\alpha$ with a parameter $F$ which is independent of time; then (7) generalizes into the Itô equation,

$$dY_t = -V'(Y_t)dt + dL_t^\alpha. \tag{9}$$

Here $L_t^\alpha$ is a symmetric $\alpha$-stable Lévy process, with $\alpha \in (0, 2)$, defined on the probability space $(\Omega, \mathcal{F}, P)$ (See Methods). The drift term $V'(Y_t)$ satisfies a Lipschitz condition with jump measures, then the solution of the SDE (9) exists and is unique (Applebaum, 2004). We focus our attention on the three THC models with different values of the parameter $F$; a weak $F$=1, fresh-water forcing $F$=1.126 which induces a symmetric potential function and a strong $F$=1.28. These values represent different geometries of a double-well potential function as in Fig. 2 (d), (e) and (f).

**2.1 Method**

This section summarizes definitions and the main properties of the stability measures used in the present analysis. Also a brief summary to the type of stochastic process chosen to model extreme events is given.





### 2.1.1 A symmetric $\alpha$-stable scalar Lévy motion

Climate extreme events show random fluctuations having sample pathways with intermittent jumps and heavy tails. Natural
and more appropriate candidate for modeling such a non-Gaussian process is an $\alpha$-stable Lévy motion. $L_t^\alpha$ with $0 < \alpha < 2$ is a stochastic process that satisfies the following properties:

a) $L_0^\alpha = 0$, almost sure;

b) $L_t^\alpha$ has independent increments;

c) stationary increments $L_t^\alpha - L_s^\alpha$ and $L_{t-s}^\alpha$ have the same symmetric $\alpha$-stable distribution, i.e.
$S_\alpha((t-s)^{\frac{1}{\alpha}}, 0, 0)$;

d) stochastically continuous sample paths, i.e., for every $s > 0$, $L_t^\alpha \to L_s^\alpha$ in probability, as $t \to s$.

The probability density function for $L_t^\alpha$ is defined by

$$t^{-\frac{1}{\alpha}} f_\alpha(t^{-\frac{1}{\alpha}} y), \tag{10}$$

where $f_\alpha$ is the probability density function for the standard symmetric $\alpha$-stable random variable $Y \sim S_\alpha(1, 0, 0)$ (for more
details see (Applebaum, 2004; Duan, 2015). The generating triplet of $L_t^\alpha$ is $(0, 0, \nu_\alpha)$, with the jump measure, i.e. the expected value of the number of jumps of size $dz$ during the unit time, is defined as:

$$\nu_\alpha(dz) = c_\alpha \frac{dz}{|z|^{1+\alpha}}, \quad \alpha \in (0, 2), \tag{11}$$

where $c_\alpha$ is the intensity constant. When $\alpha \in (0, 1)$, the $\alpha$-stable Lévy motion has finite variation, otherwise, when $\alpha \in [1, 2)$ it is unbounded.

### 2.1.2 Mean first exit time

It is defined as the first exit time from a deterministic domain $D \subset \mathbb{R}^1$ of attraction of $y_{(on/off)}$ as follows:

$$\tau(\omega, y) = \inf\{t \geq 0, Y_t(\omega, y) \notin D\}, \tag{12}$$

and the mean exit time or the mean residence time of the process in the *on(off)*-state domain is denoted as $u(y) \triangleq \mathbb{E}\tau(\omega, y) \geq 0$. It has been proven (Duan, 2015) that the mean exit time of the stochastic system (9) for an orbit starting at $y \in D$, satisfies the
following nonlocal partial differential equation with an external boundary condition

$$Au(y) = -1, \quad y \in D$$
$$u(y) = 0, \quad y \in D^c, \tag{13}$$

where $A$ is the generator defined as

$$Au(y) = -V'(y)u'(y) + \int_{\mathbb{R}^1 \setminus \{0\}} [u(y+z) - u(y) - I_{\{|z|<1\}} zu'(y)]\nu_\alpha(dz). \tag{14}$$

Here $D^c$ is the complement set of $D$ in $\mathbb{R}^1$. More over, the generator can be interpreted as $Au = \lim_{t \to 0} \frac{\mathbb{E}u(y_t) - u}{t}$, for every $u \in C^2(\mathbb{R}^1)$.





### 2.1.3 Escape probability

The likelihood that the salinity difference process $Y_t$ exits firstly from the *on(off)*-state domain $D$ by landing in the set $U \in D^c$ belonging to the *off(on)*-state domain is represented by

$$p(y) = \mathbb{P}\{Y_\tau(y) \in U\} \tag{15}$$

and solves the following integro-differential equation with the Balayage-Dirichlet boundary condition

$$Ap(y) = 0, \quad y \in D, \tag{16}$$

$$p(y) = \begin{cases} 1, & y \in U, \\ 0, & y \in D^c \backslash U. \end{cases}$$

### 2.1.4 Fokker-Planck equation

The evolution of the probability density function $p(y,t)$ for the solution paths $Y_t$ of the SDE (9) is governed by a Fokker-Planck equation

$$\partial_t p(y,t) = A^* p(y,t), \quad p(y,0) = \delta(y - y_0), \tag{17}$$

with the initial condition $Y_0 = y_0$. Where $A^*$ is the adjoint operator of $A$ (14) in the Hilbert space $L^2(\mathbb{R}^1)$ that has the explicit form in the case of a symmetric $\alpha$-stable Lévy motion. Thus, the equation (17) is specified as

$$\partial_t p(y,t) = -\partial_y \left[ f(y)p(y,t) \right] - \int_{\mathbb{R}^1 \backslash \{0\}} \left[ p(y,t) - p(y-z,t) - I_{\{|z|<1\}} \, z \, \partial_y p(y,t) \right] \nu_\alpha(dz), \tag{18}$$

where $f(y) = -V'$ is a vector field of the SDE (9).

Sufficient conditions for the existence and regularity of the probability density $p(y,t)$ in the Lévy processes driven SDE (9) is established under Hörmander's condition by using the Malliavin calculus with jump. For more details, See (Chen et.al., 2015; Song et. al., 2015; Zhang, 2014) and the references therein.

### 2.1.5 Stochastic basin of attraction

The SBA is an important theoretical and practical tool that helps to describe a metastable behavior of a system. Stochastic basin of attraction (SBA) quantifies the stability of a metastable state in a dynamical system with noise perturbation in terms of size of the basin depending on the escape probability (Serdukova et. al., 2016). SBA is the collection of initial conditions of solution processes that have low (high) probability of exit (return) from (to) a neighborhood of an attractor. This geometric tool is applicable to models with small noise and noise that is a function of an order parameter.

By Definition (Serdukova et. al., 2016): SBA of the attractor $K$ with the open deterministic domain of attraction $D$ is the set $B_K(m,M) = [\bigcup_{i=1}^{n} D_{iII}^c] \bigcup [\bigcap_{i=1}^{n} D_{iI}]$, where $D_{iI} = \{y \in D \mid p_i(y) < m\}$, $D_{iII}^c = \{y \in D_{iI}^c \mid p_i(y) > M\}$, $D_i$ are the domains of attraction of nearby attractors $K_i$ and $p(y)$ is the escape probability defined in (15).





## 3   Results and Discussion

The stability analysis of the *off* and *on*-states in the THC model is based on the three main measures: escape probability, mean first exit time and stochastic basin of attraction which are frequently used in behaviour prediction of the stochastic dynamical system trajectories and whose main properties are summarized in Section 3. In this Section we discuss the results obtained and interpret them from the climatological point of view. The potential function $V(y)$ (8) for the weak fresh-water input, as shown in Fig. 3 (a), is asymmetric and has the deepest *on*-state ($y = 0.2$) the widest $0.8$ stability basin. The length of the deterministic basin of attraction for *off*-state ($y = 1$) is $0.26$ units smaller than that for *on*-state. This indicates that under the Lévy perturbations the transition from *off*-state to *on*-state is more likely. In fact, the *off*-state basin under the noise with $\alpha = 1$ decreases by $6.75$ times, while the *on*-state basin reduces only by $1.98$. The greatest threat to the stability of *on*-state represents noise with low jumps of higher frequency, i.e. $\alpha = 1.5$. Extreme events of short time-scale (decades or less) such as human-induced greenhouse gas emission are more likely to destabilize the *on*-state. As a response to increase in greenhouse gas concentration, atmospheric water cycle boost is expected. This leads to a greater excess of precipitation over evaporation

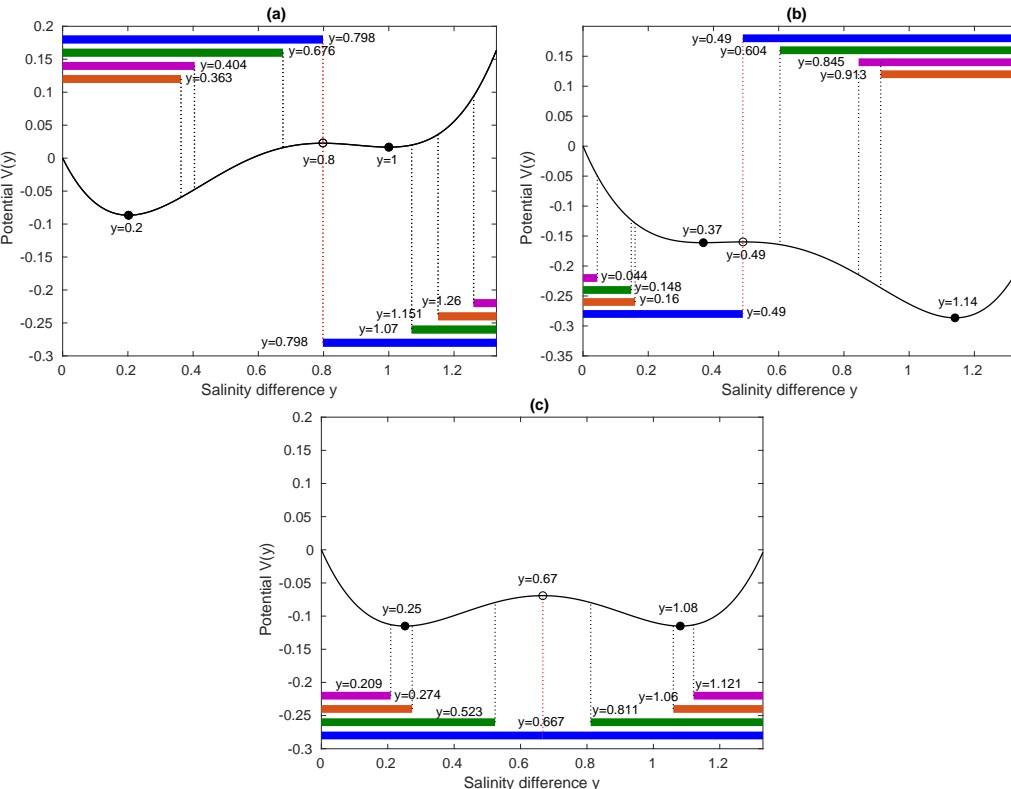

**Figure 3.** Stochastic basin of attraction for on(off)-metastable states of THC model, blue bar is a deterministic basin, green bar for $\alpha = 0.5$, violet bar for $\alpha = 1$ and orange bar for $\alpha = 1.5$. (a) $B_{0.2}(0.3, 0.8)$ and $B_1(0.3, 0.8)$ for $V(y)$ with $F = 1$; (b) $B_{0.37}(0.3, 0.8)$ and $B_{1.14}(0.3, 0.8)$ for $V(y)$ with $F = 1.28$ and (c) $B_{0.25}(0.3, 0.8)$ and $B_{1.08}(0.3, 0.8)$ for $V(y)$ with $F = 1.126$.





in high latitudes, freshening and reducing the density of surface water, and hence its tendency to sink and weaken the THC. The *off*-state stability, on the other hand, is more vulnerable to the perturbations with moderate jumps and frequencies, i.e $\alpha = 1$. Centennial-scale wind driven circulation towards higher latitudes causes a strong salting of the tropical North Atlantic

and subsequently enhances the recovery of THC.

The analysis of the trajectories' first mean exit time (12) and (13) from the deterministic basin Fig. 4(a) confirms the stability results based on the SBA.

Actually, the trajectories remain longer in the vicinity of *on*-state than in the *off*-state basin as the stability index (that is, $\alpha$) increases. The relationship between the residence time and stability index can be understood as the extreme event modeled by

Lévy motion with small jumps and high frequency ($\alpha = 1.5$) contributes to faster escape compared to the events characterized by moderate or big jumps that correspond to moderate or low probability ($\alpha = 1$ or $0.5$).

For strong fresh-water forcing the potential function $V(y)$ is also asymmetric, but now with the reversed depth: *off*-state $y = 1.14$ here is deeper than *on*-state $y = 0.37$. In this model Fig. 3 (b), *off*-state is more stable than *on*-state and this implies that under the influence of stochastic noise, the transitions from *on* to *off* are more expected.

Comparing the dimension of the effects that noise with different $\alpha$ parameter values causes in the global thermohaline circulation, it should be noted that the extreme events modeled by Lévy motion with $\alpha = 0.5$ does not reduce the stability of the *off*-state significantly, inducing only $0.114$ unit cutback in the basin width. In the case of weak THC stable state, large fresh-water outburst due to massive ice melting is rare but high impact event that would rather favour a total shutdown. The

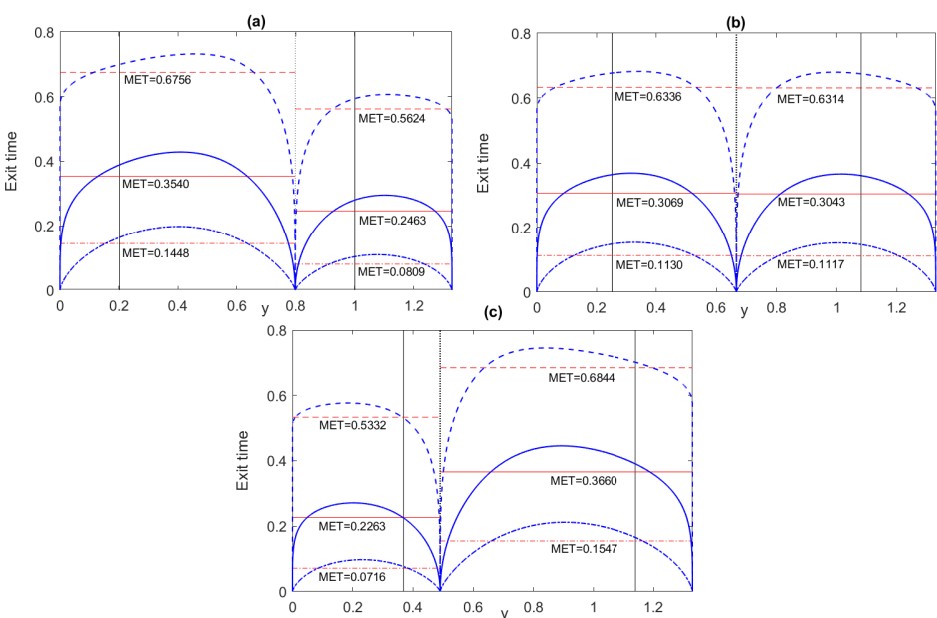

**Figure 4.** First exit time and mean first exit time MET of orbits with $\alpha = 0.5$ (dashed line), $\alpha = 1$ (solid line), $\alpha = 1.5$ (dash-dotted line) from the respective deterministic basins for (a) the potential $V(y)$ with $F = 1$; (b) the potential $V(y)$ with $F = 1.126$; (c) the potential $V(y)$ with $F = 1.28$.

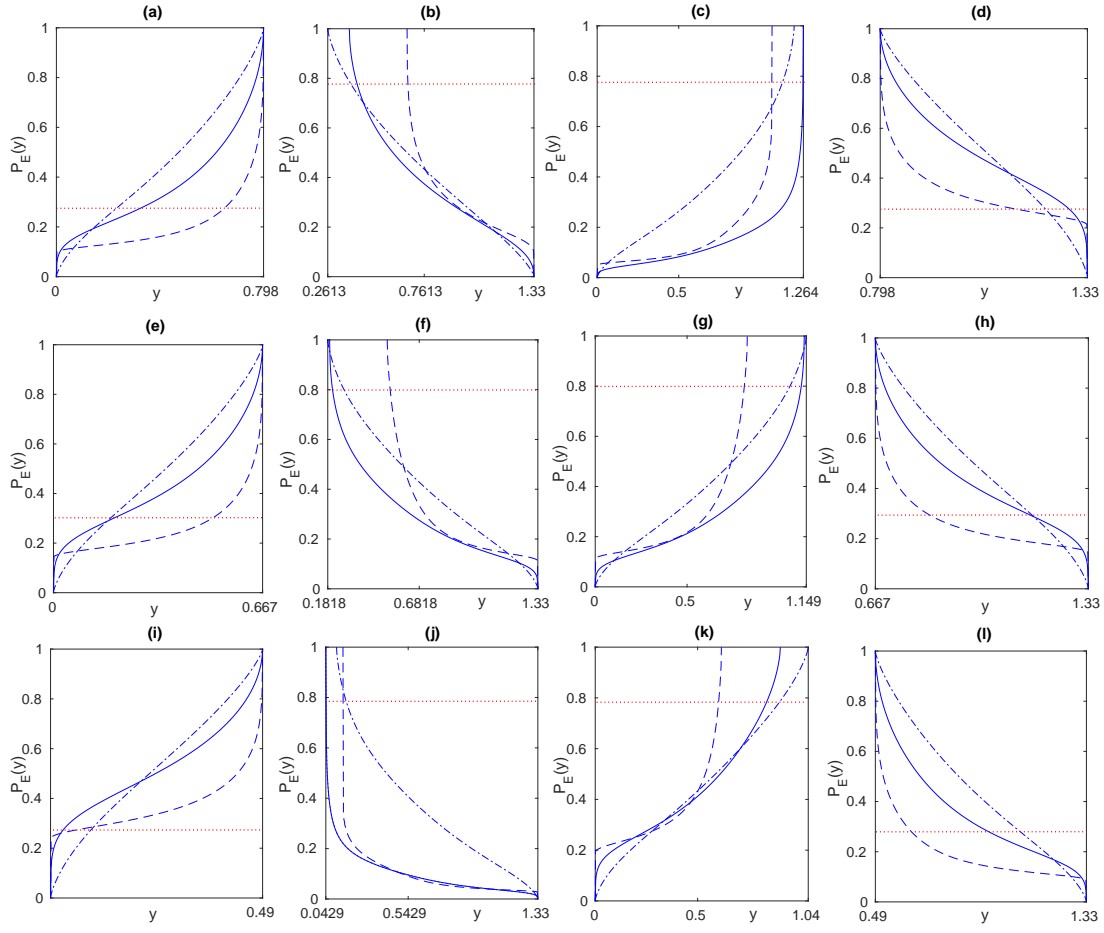

**Figure 5.** Escape probability of solutions with $\alpha = 0.5$ (dashed line), $\alpha = 1$ (solid line), $\alpha = 1.5$ (dash-dotted line) (a) from $D = (0, 0.798)$ to $D^c = (0.798, 1.33)$; (b) from $D_I^c = (0.6723, 1.33)/(0.3731, 1.33)/(0.2613, 1.33)$ to $D_I = (0, 0.6723)/(0, 0.3731)/(0, 0.2613)$ for the respective $\alpha$; (c) from $D = (0.798, 1.33)$ to $D^c = (0, 0.798)$; (d) from $D_I^c = (0, 1.072)/(0, 1.264)/(0, 1.208)$ to $D_I = (1.072, 1.33)/(1.264, 1.33)/(1.208, 1.33)$; (e) from $D = (0, 0.667)$ to $D^c = (0.667, 1.33)$; (f) from $D_I^c = (0.5069, 1.33)/(0.1918, 1.33)/(0.1818, 1.33)$ to $D_I = (0, 0.5069)/(0, 0.1918)/(0, 0.1818)$; (g) from $D = (0.667, 1.33)$ to $D^c = (0, 0.667)$; (h) from $D_I^c = (0, 0.8278)/(0, 1.138)/(0, 1.149)$ to $D_I = (0.8278, 1.33)/(1.138, 1.33)/(1.149, 1.33)$; (i) from $D = (0, 0.490)$ to $D^c = (0.490, 1.33)$; (j) from $D_I^c = (0.147, 1.33)/(0.04287, 1.33)/(0.1078, 1.33)$ to $D_I = (0, 0.147)/(0, 0.04287)/(0, 0.1078)$; (k) from $D = (0.49, 1.33)$ to $D^c = (0, 0.49)$; (l) from $D_I^c = (0, 0.616)/(0, 0.9037)/(0, 1.04)$ to $D_I = (0.616, 1.33)/(0.9037, 1.33)/(1.04, 1.33)$. (a)-(d) The potential function $V(y)$ for forcing strength $F = 1$; (e)-(h) The potential function $V(y)$ for forcing strength $F = 1.126$ (i)-(l) The potential function $V(y)$ for forcing strength $F = 1.28$. Red dotted lines $P_E(y) = 0.3$ and $P_E(y) = 0.8$ are parameters $m$ and $M$.

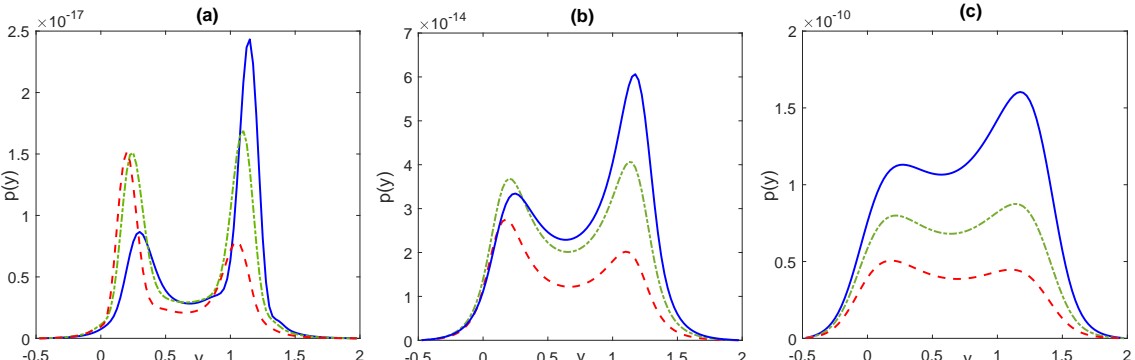

**Figure 6.** Probability density function $p(y,t)$ for the salinity difference process $Y_t$ with initial condition $Y_0 = y_0$, $t = 50$, $p(y,0) = \delta(y - y_0)$ for the potential $V(y)$ with $F = 1$ dashed orange line $y_0 = 0.8$, with $F = 1.126$ dash-dotted green line $y_0 = 0.67$, with $F = 1.28$ solid blue line $y_0 = 0.49$ and (a) $\alpha = 0.5$; (b) $\alpha = 1$; (c) $\alpha = 1.5$.

noise with the short jumps and the high frequency destabilizes the *off*-state more than the extreme events with moderate jumps
and frequency. In the case of the *on*-state stability, noise with the same parameter $\alpha = 1$ induces the greatest eleven times
shrink of its deterministic basin. Simulation results in Fig. 5 (c) indicate the noise-driven orbits escape faster from the *on*-state
and stay longer in the *off*-state basin.

The symmetry in the potential $V(y)$ is presented for $F = 1.126$ (see Fig. 3 (c)) generating the equality of the lengths of the
*on(off)*-states stability basins. Also, under the force of stochastic noise, the transition from one state to another is equiprobable.
The greatest destabilizer of the states is the perturbation with moderate jumps and probability noise, alternatively the events
characterized by low probability but high jumps bring less transient impact between the states. In Fig 4 (b), the mean exit time
of the orbits from the neighborhoods of *on(off)*-states is the same and manifests the highest values for the orbits carried by
Lévy noise with $\alpha = 0.5$. This shows as well that such type of perturbations has minimal influence on the stability of the states.

In Fig. 5 the $D_I$ and $D_{II}^C$ SBA criteria (Serdukova et. al., 2016) based on the escape probabilities (15) and (16) for the stable
*on(off)*-states of the THC model with different fresh-water forcing $F$ are shown: (a)-(d) for weak fresh-water input, (e)-(h) for
symmetric potential well inducing fresh-water input and (i)-(l) for strong forcing. The first criteria defines the set $D_I$ of the
initial salinity difference $y_0$ that originate the trajectories with the $m < 0.3$ probability of escape from the deterministic basin
of *on*-state Fig. 5 (a), (e), (i) and *off*-state (d), (h), (l). The trajectories with probability $M > 0.8$ of return to the interval $D_I$
have their initial points $y_0$ included by the second criteria in the *on*-state's SBA Fig. 5 (b), (f), (j) (*off*-state's SBA Fig. 5 (c), (g), (k)).
The geometry of the probability density functions (17) and (18) of the salinity difference process $Y_t$ Fig. 6, once again confirms
the previous conclusions about the stability of the *on(off)*-states. Therefore, the process is most likely to remain in the state
that represents the deepest valley in the potential function $V(y)$, this is in *on*-state for weak forcing and in *off*-state for strong
forcing. For symmetrical potential well inducing forcing, the orbits with equal probability visit *on* or *off*-states. The transition
between the states is more likely if the THC system undergoes to Lévy perturbations with low jumps and high probability noise
as shown in Fig. 6 (c), since the difference between the probabilities of staying in each of the states is smaller. Therefore, for
$\alpha = 0.5$ Fig. 6 (a) the probability that the process remains in *off*-state is 2.87 times greater than in *on*-state. This difference in





probabilities decreases with the increase in $\alpha$ parameter (which means a decrease in height and an increase in the frequency of jumps), resulting in a difference of $1.81$ times for $\alpha = 1$ Fig. 6 (b) and $1.42$ times for $\alpha = 1.5$ Fig. 6 (c).

## 4 Conclusion

Fluctuations in the THC patterns influence climate and civilization-induced and natural climatic changes also have impacts on THC. The THC is bistable only for some domain of the nondimensional fresh-water parameter values. Extreme events such as greenhouse gas emissions, collapse of major ice sheets and global warming induce a transition between both states. It has, therefore, great importance to sufficiently scrutinize the stability of strong (weak) THC states against these extreme events. In this work, we have performed analysis of stability of the *on*-state and *off*-state in the stochastic two-compartment thermohaline

circulation model by considering three different values of the fresh-water input control parameter.

Random noise agitations in geophysical complex dynamical systems have been regarded as continuous perturbation or Gaussian processes. However, the paleoclimatic data sets signify random fluctuations in swift climatic transitions have a non-Gaussian distribution with heavy tail and pathways that are cádlág functions with at most a countable number of jumps. Therefore, it is more fitting applying non-Gaussian symmetric $\alpha$-stable Lévy processes in modeling the influence of extreme

events in the conceptual stochastic Stommel two-compartment model. Actually, these non-Gaussian processes are becoming increasingly popular lately.

We applied three concepts, mean residence time, first passage probability and stochastic basin of attraction. Each of these quantities is helpful in understanding the stability of the strong THC (small salinity difference $y$) state and the weak THC (large salinity difference $y$) against stochastic perturbations (extreme events) and predicting transitions between these states.

The main conclusion of our work are the following: For the weak fresh-water flux, *on* to *off*-state transition is more probable under extreme events modeled by Lévy noise perturbations with smaller jumps but with high frequency. This SBA analysis result is confirmed by calculating MET of sample paths. Greenhouse gas concentration contributes to a faster escape from the strong THC state (which is analogous to the current day setting). In the deterministic case, *on*-state has the widest basin of attraction as compared to both symmetric well potential inducing and strong fresh-water influxes. Our results for strong

fresh-water input indicate extreme events of moderate jumps and frequencies facilitate the recovery of the strong THC, i.e., the transition from *off* to *on*-state is boosted. Strong THC state attraction basin suffers its largest shrink to extreme events with a moderate jump and probability Lévy noise and noise-driven pathways linger in the attraction basin of the weak THC state. Both strong and weak THC state have equal basin stability and transitions between the states are equiprobable for the symmetric potential well inducing fresh-water forcing. The geometry of the probability density function also shows that pathways shuttle

between both states. Extreme events characterized by medium jumps and frequency destabilize the states most, while events with smaller jumps and high frequency bring less transient impact.

Extreme climate events can have severe aftermaths on the ocean. Warming of the globe due to the greenhouse gas accumulation may cause magnificent melting of polar ice caps and glaciers followed by fresh-water pulses of considerable volume followed by a collapse of the THC. Total shut down of the THC is a high risk rare event with catastrophic effects on human



well-being. In this respect our results provide a useful theoretical framework for studying the oceanic thermohaline circulation under the impact of climate extreme events.

*Code availability.* Results of calculations are available by e-mail request.

*Author contributions.* DT and LS performed the computations and writing. PW and YZ contributed in computing and methods. JD and JK revised the paper.

*Competing interests.* The authors declare that they have no conflict of interest.

*Acknowledgements.* This work was partly supported by the National Natural Science Foundation of China (NSFC) (Grant Nos. 11801192, 11531006, and 11771449).





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
