# Peer review of "Influence of extreme events modeled by Lévy flight on global thermohaline circulation stability"

_Nonlinear Processes in Geophysics, 2020_

## Referee Comment (RC1) · Anonymous Referee #1 · 9 Sep 2020

Comments on npg-2020-31

The article considered influence of extreme events modeled by Lévy flight on global thermohaline circulation stability. The authors adopted the three deterministic quantitative tools: mean first passage time, escape probability and stochastic basin of attraction to character the stability of the strong THC state and the weak THC state. In my own opinion, the manuscript is clearly structured and organized. However, some points should be considered in the manuscript.

1) In the work, the symbol $V$ has two different meanings. One is the volume in Figure 1, the other is the potential function on Page 5.

2) In stochastic differential equation (9), you can add the initial condition $Y_0 = y_0$. On the second line from the bottom of Page 7, the generator should be $\lim_{t \to 0} \frac{\mathbb{E}u(y_t) - u(y_0)}{t}$.

3) In the manuscript, the authors adopted the $\alpha-$stable non-Gaussian Lévy noise to model the extreme events? Can you give the comparison between the Brownian motion and Lévy flight?

4) In equation (4), what is the definition of $I$?

5) In Section 2.1.5, what is the definition of $p_i(y), m, M$? Could you represent the definition of stochastic basin of attraction to the one-dimensional case since that the escape boundary only has two direction in the one-dimension. The work "Y. Zheng, L. Serdukova, J. Duan, J. Kurths, Transitions in a genetic transcriptional regulatory system under Lévy motion, Sci. Rep. 6 (2016) 29274." also introduces the stochastic basin of attraction, which can be added to the references.

6) In the manuscript, the authors show the three concepts, mean residence time, first passage probability and stochastic basin of attraction to perform the stability analysis. Could you show us how to solve the nonlocal equations (14) (15) and (17)?

---

## Referee Comment (RC2) · Anonymous Referee #2 · 25 Oct 2020

In this paper, the effect of $\alpha$-stable Levy noise on the transitions of the Atlantic Meridional Overturning Circulation (MOC) is studied using a variant of the Stommel two-box model. While this is an interesting topic, the paper needs a major rewrite before it can be considered for publication again. Major issues are:

a. Many of the ocean/climate statements made are incorrect. A few examples are (just on first two pages):

Title: The model does not represent the global thermohaline circulation but the Atlantic

MOC.

l17: Tides are no part of the THC.
l27: There is no surplus of precipitation over evaporation at low latitudes, except in a small zone near the equator (ITCZ).

and a full list over the whole paper would fill at least a page.

b. There should be a justification that the variability in the freshwater forcing can be represented by an $\alpha$-stable process. Here, the time scale considered is important: when focus is on Dansgaard-Oeschger (DO) events (e.g. Ditlevsen 1999), this is a different issue that when the stability of the present-day MOC is considered. As for the latter case, many observations and model results (reanalyses, CMIP6) are available for justification.

c. The new aspects in this paper, in relation to the one just published (Tesfay et al., 2020 in the reference list), should be clarified as the same model and same noise are investigated. The current paper surprisingly contains many more mistakes in formulation than in the published one, i.e.

l72: $\Delta\rho$ should be divided by $\rho_0$.
l99: $\beta$ is no restoration 'tensility' but a ratio of a diffusive and a restoring time scale
l101: definition of $\mu^2$ is wrong.
l105, 107: $dt \to d\tau$.
l129: the relation between the amplitude of $dL_t$ and $F$ is missing.
Fig. 6 contains no probability distributions as for each curve the integral is not 1.

so please correct all these (and many more).

d. The methodology in section 2.1 should be better explained and only provide well explained mathematical results with reference to the mathematical details. It appears now to have been copied from a mathematics paper with many symbols unexplained. At line 130, there is a reference to a 'Methods' section which is not there.

e. Section 3: I would suggest to split the results into two sections: (i) DO transitions. Connect the results to the Ditlevsen (1999) analysis and proposed noise structure. Can the $\alpha$-stable noise better describe the transition behavior (as in the proxy data), than just Brownian noise? (ii) Present-day MOC. Is the transition probability of a MOC transition increased under climate change, when incorporating an $\alpha$-stable process in the freshwater flux noise?

Improve also the interpretation of the results: in the present text, lines 209-210, lines 222-223, lines 267-268 and lines 277-281 make no sense.

---

## Author Comment (AC1) · 10 Jan 2021

We would like to express gratitude to the Referees for the detailed and precise analysis of our manuscript that contributes towards its improvement. We have taken all of the comments into account in the revision (changes appear in red type) and explain this in detail in the following sections.

**1 Technical comments from Reviewer 1**

**Question 1**: In the work, the symbol $V$ has two different meanings. One is the volume in Figure 1, the other is the potential function on Page 5.

**Answer**: The volume of the compartments in Figure 1 was designated as $V_c$.

**Question 2**: In stochastic differential equation (9), you can add the initial condition $Y_0 = y_0$. On the second line from the bottom of Page 7, the generator should be $Au = \lim_{t \to 0} \frac{\mathbb{E}u(y_t) - u(y_0)}{t}$.

**Answer**: The initial condition $Y_0 = y_0$ is added to the equation (9). The generator on line 165 is corrected as suggested.

**Question 3**: In the manuscript, the authors adopted the $\alpha$stable non-Gaussian Lévy noise to model the extreme events? Can you give the comparison between the Brownian motion and Lévy flight?

**Answer**: We compare the processes in the end of the section 2.1.1.

**Question 4**: In equation (14), what is the definition of $I$?

**Answer**: The definition of indicator function $I$ is given by equation (15).

**Question 5**: In Section 2.1.5, what is the definition of $p_i(y), m, M$? Could you represent the definition of stochastic basin of attraction to the one-dimensional case since that the escape boundary only has two direction in the one-dimension. The work "Y. Zheng, L. Serdukova, J. Duan, J. Kurths, Transitions in a genetic transcriptional regulatory system under Lévy motion, Sci. Rep. 6 (2016) 29274." also introduces the stochastic basin of attraction, which can be added to the references.

**Answer**: In Section 2.1.5 the definition of stochastic basin of attraction is adapted to the one-dimensional case and the measures of $m$ and $M$ are specified. The work of Y. Zheng is added to the references line 355.

**Question 6**: In the manuscript, the authors show the three concepts, mean residence time, first passage probability and stochastic basin of attraction to perform the stability analysis. Could you show us how to solve the non local equations (14) (15) and (17)?

**Answer**: At the end of section 2.1.3 we describe the numerical method that we use to solve these equations.

**2 Technical comments from Reviewer 2**

**Question 1**: Title: The model does not represent the global thermohaline circulation but the Atlantic MOC.

**Answer**: We have made proposed amendments to both the title and the text.

**Question 2**: l17: Tides are no part of the THC. l27: There is no surplus of precipitation over evaporation at low latitudes, except in a small zone near the equator (ITCZ).

**Answer**: The suggested changes are introduced in the lines 16 and 26.

**Question 3**: There should be a justification that the variability in the freshwater forcing can be represented by an $\alpha$-stable process. Here, the time scale considered is important: when focus is on Dansgaard-Oeschger (DO) events (e.g. Ditlevsen 1999), this is a different issue that when the stability of the present-day MOC is considered. As for the latter case, many observations and model results (reanalyses, CMIP6) are available for justification.

**Answer**: We consult the publications (bibliography line 372) and introduce the suggested justification on lines 55-60.

**Question 4**: The new aspects in this paper, in relation to the one just published (Tesfay et al., 2020 in the reference list), should be clarified as the same model and same noise are investigated.

**Answer**: This clarification is included on page 3 in the first paragraph.

**Question 5**: l72: $\Delta\rho$ should be divided by $\rho_0$. l99: $\beta$ is no restoration "tensility" but a ratio of a diffusive and a restoring time scale. l101: definition of $\mu^2$ is wrong. l105, 107: $dt \to d\tau$. l129: the relation between the amplitude of $dL_t$ and $F$ is missing.

**Answer**: The respective corrections were introduced in the equations (2), (6) and (7).

**Question 6**: Fig. 6 contains no probability distributions as for each curve the integral is not 1.

**Answer**: Figure 6 has been replaced according to the suggestion.

**Question 7**: The methodology in section 2.1 should be better explained and only provide well explained mathematical results with reference to the mathematical details. It appears now to have been copied from a mathematics paper with many symbols unexplained. At line 130, there is a reference to a "Methods" section which is not there.

**Answer**: All mathematical concepts were described in more detail in section 3, on pages 7, 8 and 9.

**Question 8**: Section 3: I would suggest to split the results into two sections: (i) DO transitions. Connect the results to the Ditlevsen (1999) analysis and proposed noise structure. Can the $\alpha$-stable noise better describe the transition behavior (as in the proxy data), than just Brownian noise? (ii) Present-day MOC. Is the transition probability of a MOC transition increased under climate change, when incorporating an $\alpha$-stable process in the freshwater flux noise?

**Answer**: To make the proposed comparison ($\alpha$-stable vs Brownian noise) we should include the parameter $\alpha = 2$ (which corresponds to the Brownian case) in the simulations of stochastic perturbations. We leave this option for future studies.

**Question 9**: Improve also the interpretation of the results: in the present text, lines 209-210, lines 222-223, lines 267-268 and lines 277-281 make no sense.

**Answer**: Reading more articles about timescales of AMOC decline, AMOC response to fresh water forcing and stability of AMOC off-state we try to improve the interpretation of the results, see the changes

made in the section 4.

---

## Author Comment (AC2) · 10 Jan 2021

The comment was uploaded in the form of a supplement:
https://npg.copernicus.org/preprints/npg-2020-31/npg-2020-31-AC2-supplement.pdf

---

## Author Comment (AC3) · 10 Jan 2021

**Influence of extreme events modeled by Lévy flight on Atlantic meridional overturning circulation stability**

Daniel Tesfay[1], Larissa Serdukova[2], Yayun Zheng[1], Pingyuan Wei[1], Jinqiao Duan[3], and Jürgen Kurths[4]

[1]School of Mathematics and Statistics & Wuhan Center for Mathematical Sciences, Huazhong University of Science and Technology, Wuhan 430074, China
[2]Department of Mathematics and Statistics, Reading University, Reading RG6 6AX, UK
[3]Department of Applied Mathematics, Illinois Institute of Technology, 312-567-5335, Chicago 60616, USA
[4]Research Domain on Transdisciplinary Concepts and Methods, Potsdam Institute for Climate Impact Research, PO Box 60 12 03, 14412 Potsdam, Germany

**Correspondence:** Larissa Serdukova (l.i.serdukova@reading.ac.uk)

**Abstract.**

How will extreme events due to human activities and climate change affect the Atlantic meridional overturning circulation is a key concern in climate predictions. The stability of the thermohaline circulation with respect to extreme events, such as fresh-water oscillations is examined using a conceptual stochastic Stommel two-compartment model. The extreme fluctuations are modeled by symmetric $\alpha$-stable Lévy motions whose pathways are cádlág functions with at most a countable number of jumps. The mean first passage time, escape probability and stochastic basin of attraction are used to perform the stability analysis of *on (off)* equilibrium states. Our results argue that for model with weak fresh-water forcing strength, the greatest threat to the stability of the *on*-state represents noise with low jumps and higher frequency that can be seen as freshwater inputs from glacier melting due to ocean warming caused by increased greenhouse gas emissions. On the other hand, the *off*-state stability is more vulnerable to the agitations with moderate jumps and frequencies which can be interpreted as a possible scenario of Atlantic thermohaline circulation recovery. Under the repercussion of stochastic noise, *on* to *off* transitions are more expected in the model with the strong fresh-water influx. Moreover, transitions from one metastable state to another are equiprobable when the fresh-water input induces a symmetric potential well.

**1 Introduction**

Natural and civilization catalyzed fluctuations in climate have significant impact on the ocean and ocean circulation pattern variations greatly affect climate (Chapman and Shackleton, 1999; Clark et. al., 2002). The thermohaline circulation, known as great ocean conveyor as well, has been declared potentially unstable, whose change could lead to abrupt climate shift on all timescales (Marotzke, 2000). Thermohaline circulation is basically an outcome of the interplay of fresh-water with thermal energy along with the ocean-atmosphere interface and inside the ocean competition of temperature and salinity (Rahmstorf, 2003). This enormous oceanic process has a significant contribution in maintaining the equilibrium of Earth's energy framework by redistributing thermal energy of the order $1.2 \times 10^{15}$W northwards in the Atlantic ocean. A large proportion of the

meridional overturning circulation (MOC) is usually categorized as thermohaline circulation because MOC takes the lion's share in this heat penetration to the north pole (Ganachaud and Wunsch, 2001; Trenberth and Solomon, 1994). The warm and saltier surface water on interannual and even longer time-scales gets freshened and loses heat to the cold atmosphere. Subsequently, the water descends slightly to the bottom of the Atlantic since it gets denser than the underneath water. The cooled water eventually returns southward as deep current and the warm temperature around the equatorial belt opts for upwelling. Thermohaline circulation is a combination of the floating of deep water currents around the equator and in the southern oceans, the horizontal currents, and the descending and forming of deep water in high latitudes. Climate reconstructions indicate that conveyor belt has indispensable contribution in the climate system transitions from cold to warm or from warm to cold climate states (Rahmstorf, 1995; Bond et.al., 1997; Grootes and Stuiver, 1997). In today's warm climate, the thermohaline circulation state is sensitive to increased fresh-water volume. During global warming, the intensity of the water cycle, especially at high latitudes, increases, and the melting of ice accelerates, the supply of fresh water to the North Atlantic will most likely increase, thereby reducing the density of the surface layer in the MOC sinking area. Thus, it seems likely that the thermohaline circulation will weaken over the coming century. Studying the stability of this oceanic conveyor belt by analyzing the influence of internal and external agitations on its dynamical behavior is increasingly pressing presently.

To study how the Atlantic meridional overturning circulation (AMOC) transports properties latitudinally, a conceptual deterministic two-compartment model was forwarded by Stommel (Stommel, 1961). Shreds of evidence from sea observation and model simulation show that the strength of the thermohaline flow is sensitive to the surface fresh-water flux fluctuation (Jackson and Wood, 2017; Caesar et.al., 2018). Competition between thermal versus saline forcing can lead to a multiple equilibria regime if the relaxation time-scales for the temperature and the salinity are distinct. The thermohaline flow system is bistable, one with strong circulation (analogous with the present set up), and the second state with a very weak flow, when the salinity difference is forced by a prescribed fresh-water flux (Marotzke, 2000). The multistability of AMOC is also verified by results obtained from different numerical models (Broecker, 1987). The deterministic compartment model has been further extended to include noisy thermal and saline forcing oscillations (Huang et. al., 1992; Rahmstorf, 1996; Djikstra, 2005).

Forcing the general ocean circulation model with some particularly large stochastic fresh-water fluctuations is found to trigger pulsation of transport from one stable configuration to the other (Mikolajewicz and Maier-Reimer, 1990). Therefore, the fresh-water budget is the main control parameter of the Atlantic circulation, the oscillations of which lead to the circulation response in the form of a hysteresis curve. There is evidence that after the AMOC crossing a threshold exists a temporary resilience period during which the AMOC could still recover if freshwater inflow ceases (Jackson and Wood, 2017).

The Gaussian noise perturbed thermohaline circulation has been under extensive study. For instance, it was shown in (Vélez-Belchí et. al., 2001) that an increment of 5% of the fresh-water forcing in the ocean circulation could stimulate transitions between a high and low salinity difference metastable states. In a time-dependent compartment model for thermohaline circulation with Brownian motion and moderate noise intensity, hysteresis does not adiabatically follow stationary distribution (Bergund and Gentz, 2002). Meanwhile, the noise forcing climate comprises of a non-Gaussian $\alpha$-stable Lévy noise component (Fuhrer et. al., 1993; Ditlevsen, 1999). The occurrence more than a dozen of additional Dansgaard-Oeschger (D-O) events that took place during the last glacial period could not have been reproduced by using the continuous perturbation processes. As

well modern climate models from the fifth Coupled Model Intercomparison Project (CMIP5) predict abrupt non-linear shifts in subpolar North Atlantic dynamics (Sgubin et. al., 2016). The jumps in those events could better be modeled by Lévy flights (Kuhwald and Pavlyukevich, 2016).

60      Paleoclimatic data indicate the coincidence of transitions from strong to weak or from weak to strong thermohaline circulation states with the occurrence of extreme climatic variations (Vélez-Belchí et. al., 2001). In our previous work (Tesfay et. al., 2020), we investigated the most probable trajectories of such transitions. The results of this study led us to a number of questions, of how is the overturning circulation stable to abrupt climatic changes? And what parameters of Lévy noise most affect the equilibrium of the AMOC? We also estimate possible scenarios for thermohaline circulation regeneration, analyzing

65 the stability of the *off* state under stochastic fresh-water fluctuations.

     We analyze the stability of metastable states of the AMOC model by calculating three quantities, namely, mean first passage (exit) time, first passage (escape) probability and stochastic basin of attraction that carry the dynamical information of the model. In the present form of the AMOC, analyzing the intensity and mechanism of the forcing schemes that could trigger such transitions and studying the stability of strong and weak AMOC equilibrium states is of fundamental importance.

70      Particularly, we will study the effect of extreme events on the scalar stochastic AMOC model

$$dY_t = -V'(Y_t)dt + dL_t^\alpha \tag{1}$$

by measuring the stability of equilibrium states of the salinity difference process $Y_t$ for various values of (nondimensional) fresh-water forcing and non-Gaussianity parameter $\alpha$. In Eq. (1), $V$ is a potential function (details are given in Section 2).

     The paper is structured as follows. In Section 2, we discuss the simplified conceptual stochastic Stommel two-compartment

[revised manuscript text omitted]

The original system (4) with the substitutions $\quad x \equiv \frac{\Delta T}{\theta}, \quad y \equiv \frac{\Delta S\beta_s}{\theta\beta_T}, \quad \tau \equiv \frac{t}{t_d}$ is reduced to the dimensionless system of evolution equations (Cessi, 1994; Tesfay et. al., 2020)

$$dx = (-\beta(x-1) - x[1 + \mu^2(x-y)^2])d\tau,$$
$$dy = (F - y[1 + \mu^2(x-y)^2])d\tau, \tag{5}$$

where $\beta$ a ratio of a diffusive and a restoring time scale, $\mu^2$ strength of the buoyancy-driven convection between the two compartments relative to the diffusive mixing and $F$ dimensionless fresh-water forcing are defined as

$$\beta = \frac{t_d}{t_r}, \qquad \mu^2 = \frac{qt_d(\beta_T\theta)^2}{V}, \qquad F = \frac{\beta_S S_0 t_d}{\beta_T\theta H}F_S. \tag{6}$$

[revised manuscript text omitted]

$$\nu_\alpha(dz) = \frac{\alpha \, \Gamma((1+\alpha)/2) \, dz}{2^{1-\alpha} \sqrt{\pi} \, \Gamma(1-\alpha/2) \, |z|^{1+\alpha}}, \quad \alpha \in (0, 2), \tag{11}$$

where $\Gamma$ is the Gamma function. When $\alpha \in (0,1)$, the $\alpha$-stable Lévy motion has finite variation, otherwise, when $\alpha \in [1,2)$ it is unbounded.

Comparing two stochastic processes such as Brownian motion and Lévy flight, several differences and some similarities can be distinguished. This comparison will once again justify our choice of noise for simulating climate extreme events.

Both processes are a random walk with independent and stationary increments. Brownian motion increments have a Normal distribution, while Lévy flight step-lengths have a Lévy distribution, a probability distribution that is heavy-tailed. The sample path of the Brownian motion is continuous, differing in this from Lévy flight since its path is a cadlag function. Namely, Lévy path at most has a countable number of jumps, which are the only discontinuities in time.

**3.2 Mean first exit time**

The first exit time for a solution orbit from a deterministic domain $D \subset \mathbb{R}^1$ of attraction of $y_{(on/off)}$ is defined as:

$$\tau(\omega, y) = \inf\{t \geq 0, Y_t(\omega, y) \notin D\}, \tag{12}$$

and the mean exit time or the mean residence time of the process in the *on(off)*-state domain is denoted as $u(y) \triangleq \mathbb{E}\tau(\omega, y) \geq 0$. It has been proven (Duan, 2015) that the mean exit time of the stochastic system (9) for an orbit starting at $y \in D$, satisfies the following nonlocal partial differential equation with an external boundary condition

$$Au(y) = -1, \quad y \in D$$
$$u(y) = 0, \quad y \in D^c, \tag{13}$$

where $A$ is the generator defined as

$$Au(y) = -V'(y)u'(y) + \int_{\mathbb{R}^1\backslash\{0\}} [u(y+z) - u(y) - I_{\{|z|<1\}}\, zu'(y)]\nu_\alpha(dz). \tag{14}$$

Here $D^c$ is the complement set of $D$ in $\mathbb{R}^1$ and $I_{\{|z|<1\}}(z)$ is the indicator function for a set $|z| < 1$, defined as

$$I_{\{|z|<1\}}(z) = \begin{cases} 1, & \text{if } |z| < 1, \\ 0, & \text{if } |z| \geq 1. \end{cases} \tag{15}$$

Moreover, the generator can be interpreted as $Au = \lim\limits_{t \to 0} \frac{\mathbb{E}u(y_t) - u(y_0)}{t}$, for every $u \in C^2(\mathbb{R}^1)$.

**3.3 Escape probability**

The likelihood that the salinity difference process $Y_t$ exits firstly from the *on(off)*-state domain $D$ by landing in the set $U \in D^c$ belonging to the *off(on)*-state domain is represented by

$$p(y) = \mathbb{P}\{Y_\tau(y) \in U\} \tag{16}$$

and solves the following integro-differential equation with the Balayage-Dirichlet boundary condition

$$Ap(y) = 0, \quad y \in D, \tag{17}$$

$$p(y) = \begin{cases} 1, & y \in U, \\ 0, & y \in D^c\backslash U. \end{cases}$$

We use a numerical approach adapted from Gao et. al. (2014) for solving equation (17). The simplified form of this equation

$$\frac{d}{2}p''(x) + f(x)p'(x) - \frac{\varepsilon C_\alpha}{\alpha}[\frac{1}{(1+x)^\alpha} + \frac{1}{(1-x)^\alpha}]p(x)$$

$$+\varepsilon C_\alpha \int_{-1-x}^{1-x} \frac{p(x+y) - p(x)}{|y|^{1+\alpha}}dy = 0, \tag{18}$$

is discretized in the $x \in (-1, 1)$. $p(x) = 1$ for $x \in (1, +\infty)$ and $p(x) = 0$ for $x \in (-\infty, -1)$. The corresponding schemes for the general case can be extended easily, using central difference for derivatives and the "punched-hole" trapezoidal rule getting a discretized equation of second-order accuracy for any $0 < \alpha < 2$ and $j = -J+1, ..., -2, -1, 0, 1, 2, ..., J-1$:

$$
C_h \frac{P_{j-1} - 2P_j + P_{j+1}}{h^2} + f(x_j) \frac{P_{j+1} - P_{j-1}}{2h}
$$

$$
- \frac{\varepsilon C_{Pj}}{\alpha} \left[ \frac{1}{(1+x_j)^\alpha} + \frac{1}{(1-x_j)^\alpha} \right] + \varepsilon C_\alpha h \sum_{k=-J-j, k \neq 0}^{J-j} {}'' \frac{P_{j+k} - P_j}{|x_k|^{1+\alpha}} = 0 \tag{19}
$$

where $P_j$ is the numerical solution of $p$ at $x_j$. The interval $[-2, 2]$ is divided into $4J$ subintervals and $x_j = jh$ for $-2J \leq j \leq 2J$ integers, where $h = 1/J$. The modified summation symbol $\sum''$ means that the quantities corresponding to the two end summation indices are multiplied by $1/2$. For more information see Gao et. al. (2014).

**3.4 Fokker-Planck equation**

[revised manuscript text omitted]

Lévy motion with small jumps and high frequency ($\alpha = 1.5$) contributes to faster escape compared to the events characterized by moderate or big jumps that correspond to moderate or low probability ($\alpha = 1$ or $0.5$). Actually, when rate of fresh-water released is large (Jackson and Wood, 2017) the timescale of AMOC weakening is measured in decades and does not depend on the actual rate of fresh-water inflow, because advective feedbacks become enhanced.

For strong fresh-water forcing the potential function $V(y)$ is also asymmetric, but now with the reversed depth: *off*-state $y = 1.14$ here is deeper than *on*-state $y = 0.37$. In this model Fig. 3 (b), *off*-state is more stable than *on*-state and this implies that under the influence of stochastic noise, the transitions from *on* to *off* are more expected.

Comparing the dimension of the effects that noise with different $\alpha$ parameter values causes in the meridional overturning circulation, it should be noted that the extreme events modeled by Lévy motion with $\alpha = 0.5$ does not reduce the stability of the *off*-state significantly, inducing only $0.114$ unit cutback in the basin width. The noise with the short jumps and the high frequency destabilizes the *off*-state more than the extreme events with moderate jumps and frequency. The long-term stability (450-year duration of the model integration) of AMOC *off*-state was also revealed in the study of eddy-permitting climate model (Mecking et. al., 2016) and explained by the combination of the anomalous northward freshwater transport with the freshening due to reduced evaporation in this region.

[revised manuscript text omitted]

---

## Author Comment (AC4) · 10 Jan 2021

We would like to express gratitude to the Referees for the detailed and precise analysis of our manuscript that contributes towards its improvement. We have taken all of the comments into account in the revision (changes appear in red type) and explain this in detail in the following sections.

**1    Technical comments from Reviewer 1**

**Question 1**: In the work, the symbol $V$ has two different meanings. One is the volume in Figure 1, the other is the potential function on Page 5.

**Answer**: The volume of the compartments in Figure 1 was designated as $V_c$.

**Question 2**: In stochastic differential equation (9), you can add the initial condition $Y_0 = y_0$. On the second line from the bottom of Page 7, the generator should be $Au = \lim_{t \to 0} \frac{\mathbb{E}u(y_t) - u(y_0)}{t}$.

**Answer**: The initial condition $Y_0 = y_0$ is added to the equation (9). The generator on line 165 is corrected as suggested.

**Question 3**: In the manuscript, the authors adopted the $\alpha$stable non-Gaussian Lévy noise to model the extreme events? Can you give the comparison between the Brownian motion and Lévy flight?

**Answer**: We compare the processes in the end of the section 2.1.1.

**Question 4**: In equation (14), what is the definition of $I$?

**Answer**: The definition of indicator function $I$ is given by equation (15).

**Question 5**: In Section 2.1.5, what is the definition of $p_i(y), m, M$? Could you represent the definition of stochastic basin of attraction to the one-dimensional case since that the escape boundary only has two direction in the one-dimension. The work "Y. Zheng, L. Serdukova, J. Duan, J. Kurths, Transitions in a genetic transcriptional regulatory system under Lévy motion, Sci. Rep. 6 (2016) 29274." also introduces the stochastic basin of attraction, which can be added to the references.

**Answer**: In Section 2.1.5 the definition of stochastic basin of attraction is adapted to the one-dimensional case and the measures of $m$ and $M$ are specified. The work of Y. Zheng is added to the references line 355.

**Question 6**: In the manuscript, the authors show the three concepts, mean residence time, first passage probability and stochastic basin of attraction to perform the stability analysis. Could you show us how to solve the non local equations (14) (15) and (17)?

**Answer**: At the end of section 2.1.3 we describe the numerical method that we use to solve these equations.

**2    Technical comments from Reviewer 2**

**Question 1**: Title: The model does not represent the global thermohaline circulation but the Atlantic MOC.

**Answer**: We have made proposed amendments to both the title and the text.

**Question 2**: l17: Tides are no part of the THC. l27: There is no surplus of precipitation over evaporation at low latitudes, except in a small zone near the equator (ITCZ).

**Answer**: The suggested changes are introduced in the lines 16 and 26.

**Question 3**:There should be a justification that the variability in the freshwater forcing can be represented by an $\alpha$-stable process. Here, the time scale considered is important: when focus is on Dansgaard-Oeschger (DO) events (e.g. Ditlevsen 1999), this is a different issue that when the stability of the present-day MOC is considered. As for the latter case, many observations and model results (reanalyses, CMIP6) are available for justification.

**Answer**: We consult the publications (bibliography line 372) and introduce the suggested justification on lines 55-60.

**Question 4**: The new aspects in this paper, in relation to the one just published (Tesfay et al., 2020 in the reference list), should be clarified as the same model and same noise are investigated.

**Answer**: This clarification is included on page 3 in the first paragraph.

**Question 5**: l72: $\Delta\rho$ should be divided by $\rho_0$. l99: $\beta$ is no restoration "tensility" but a ratio of a diffusive and a restoring time scale. l101: definition of $\mu^2$ is wrong. l105, 107: $dt \to d\tau$. l129: the relation between the amplitude of $dL_t$ and $F$ is missing.

**Answer**: The respective corrections were introduced in the equations (2), (6) and (7).

**Question 6**: Fig. 6 contains no probability distributions as for each curve the integral is not 1.

**Answer**: Figure 6 has been replaced according to the suggestion.

**Question 7**: The methodology in section 2.1 should be better explained and only provide well explained mathematical results with reference to the mathematical details. It appears now to have been copied from a mathematics paper with many symbols unexplained. At line 130, there is a reference to a "Methods" section which is not there.

**Answer**: All mathematical concepts were described in more detail in section 3, on pages 7, 8 and 9.

**Question 8**: Section 3: I would suggest to split the results into two sections: (i) DO transitions. Connect the results to the Ditlevsen (1999) analysis and proposed noise structure. Can the $\alpha$-stable noise better describe the transition behavior (as in the proxy data), than just Brownian noise? (ii) Present-day MOC. Is the transition probability of a MOC transition increased under climate change, when incorporating an $\alpha$-stable process in the freshwater flux noise?

**Answer**: To make the proposed comparison ($\alpha$-stable vs Brownian noise) we should include the parameter $\alpha = 2$ (which corresponds to the Brownian case) in the simulations of stochastic perturbations. We leave this option for future studies.

**Question 9**: Improve also the interpretation of the results: in the present text, lines 209-210, lines 222-223, lines 267-268 and lines 277-281 make no sense.

**Answer**: Reading more articles about timescales of AMOC decline, AMOC response to fresh water forcing and stability of AMOC off-state we try to improve the interpretation of the results, see the changes

made in the section 4.

**Influence of extreme events modeled by Lévy flight on Atlantic meridional overturning circulation stability**

Daniel Tesfay[1], Larissa Serdukova[2], Yayun Zheng[1], Pingyuan Wei[1], Jinqiao Duan[3], and Jürgen Kurths[4]

[1]School of Mathematics and Statistics & Wuhan Center for Mathematical Sciences, Huazhong University of Science and Technology, Wuhan 430074, China
[2]Department of Mathematics and Statistics, Reading University, Reading RG6 6AX, UK
[3]Department of Applied Mathematics, Illinois Institute of Technology, 312-567-5335, Chicago 60616, USA
[4]Research Domain on Transdisciplinary Concepts and Methods, Potsdam Institute for Climate Impact Research, PO Box 60 12 03, 14412 Potsdam, Germany

**Correspondence:** Larissa Serdukova (l.i.serdukova@reading.ac.uk)

**Abstract.**

How will extreme events due to human activities and climate change affect the Atlantic meridional overturning circulation is a key concern in climate predictions. The stability of the thermohaline circulation with respect to extreme events, such as fresh-water oscillations is examined using a conceptual stochastic Stommel two-compartment model. The extreme fluctuations are modeled by symmetric $\alpha$-stable Lévy motions whose pathways are cádlág functions with at most a countable number of jumps. The mean first passage time, escape probability and stochastic basin of attraction are used to perform the stability analysis of *on (off)* equilibrium states. Our results argue that for model with weak fresh-water forcing strength, the greatest threat to the stability of the *on*-state represents noise with low jumps and higher frequency that can be seen as freshwater inputs from glacier melting due to ocean warming caused by increased greenhouse gas emissions. On the other hand, the *off*-state stability is more vulnerable to the agitations with moderate jumps and frequencies which can be interpreted as a possible scenario of Atlantic thermohaline circulation recovery. Under the repercussion of stochastic noise, *on* to *off* transitions are more expected in the model with the strong fresh-water influx. Moreover, transitions from one metastable state to another are equiprobable when the fresh-water input induces a symmetric potential well.

**1 Introduction**

Natural and civilization catalyzed fluctuations in climate have significant impact on the ocean and ocean circulation pattern variations greatly affect climate (Chapman and Shackleton, 1999; Clark et. al., 2002). The thermohaline circulation, known as great ocean conveyor as well, has been declared potentially unstable, whose change could lead to abrupt climate shift on all timescales (Marotzke, 2000). Thermohaline circulation is basically an outcome of the interplay of fresh-water with thermal energy along with the ocean-atmosphere interface and inside the ocean competition of temperature and salinity (Rahmstorf, 2003). This enormous oceanic process has a significant contribution in maintaining the equilibrium of Earth's energy framework by redistributing thermal energy of the order $1.2 \times 10^{15}$W northwards in the Atlantic ocean. A large proportion of the

meridional overturning circulation (MOC) is usually categorized as thermohaline circulation because MOC takes the lion's share in this heat penetration to the north pole (Ganachaud and Wunsch, 2001; Trenberth and Solomon, 1994). The warm and saltier surface water on interannual and even longer time-scales gets freshened and loses heat to the cold atmosphere. Subse-
25  quently, the water descends slightly to the bottom of the Atlantic since it gets denser than the underneath water. The cooled water eventually returns southward as deep current and the warm temperature around the equatorial belt opts for upwelling. Thermohaline circulation is a combination of the floating of deep water currents around the equator and in the southern oceans, the horizontal currents, and the descending and forming of deep water in high latitudes. Climate reconstructions indicate that conveyor belt has indispensable contribution in the climate system transitions from cold to warm or from warm to cold climate
30  states (Rahmstorf, 1995; Bond et.al., 1997; Grootes and Stuiver, 1997). In today's warm climate, the thermohaline circulation state is sensitive to increased fresh-water volume. During global warming, the intensity of the water cycle, especially at high latitudes, increases, and the melting of ice accelerates, the supply of fresh water to the North Atlantic will most likely increase, thereby reducing the density of the surface layer in the MOC sinking area. Thus, it seems likely that the thermohaline circulation will weaken over the coming century. Studying the stability of this oceanic conveyor belt by analyzing the influence of
35  internal and external agitations on its dynamical behavior is increasingly pressing presently.

To study how the Atlantic meridional overturning circulation (AMOC) transports properties latitudinally, a conceptual deterministic two-compartment model was forwarded by Stommel (Stommel, 1961). Shreds of evidence from sea observation and model simulation show that the strength of the thermohaline flow is sensitive to the surface fresh-water flux fluctuation (Jackson and Wood, 2017; Caesar et.al., 2018). Competition between thermal versus saline forcing can lead to a multiple equilibria
40  regime if the relaxation time-scales for the temperature and the salinity are distinct. The thermohaline flow system is bistable, one with strong circulation (analogous with the present set up), and the second state with a very weak flow, when the salinity difference is forced by a prescribed fresh-water flux (Marotzke, 2000). The multistability of AMOC is also verified by results obtained from different numerical models (Broecker, 1987). The deterministic compartment model has been further extended to include noisy thermal and saline forcing oscillations (Huang et. al., 1992; Rahmstorf, 1996; Djikstra, 2005).

45  Forcing the general ocean circulation model with some particularly large stochastic fresh-water fluctuations is found to trigger pulsation of transport from one stable configuration to the other (Mikolajewicz and Maier-Reimer, 1990). Therefore, the fresh-water budget is the main control parameter of the Atlantic circulation, the oscillations of which lead to the circulation response in the form of a hysteresis curve. There is evidence that after the AMOC crossing a threshold exists a temporary resilience period during which the AMOC could still recover if freshwater inflow ceases (Jackson and Wood, 2017).

50  The Gaussian noise perturbed thermohaline circulation has been under extensive study. For instance, it was shown in (Vélez-Belchí et. al., 2001) that an increment of 5% of the fresh-water forcing in the ocean circulation could stimulate transitions between a high and low salinity difference metastable states. In a time-dependent compartment model for thermohaline circulation with Brownian motion and moderate noise intensity, hysteresis does not adiabatically follow stationary distribution (Bergund and Gentz, 2002). Meanwhile, the noise forcing climate comprises of a non-Gaussian $\alpha$-stable Lévy noise component
55  (Fuhrer et. al., 1993; Ditlevsen, 1999). The occurrence more than a dozen of additional Dansgaard-Oeschger (D-O) events that took place during the last glacial period could not have been reproduced by using the continuous perturbation processes. As

well modern climate models from the fifth Coupled Model Intercomparison Project (CMIP5) predict abrupt non-linear shifts in subpolar North Atlantic dynamics (Sgubin et. al., 2016). The jumps in those events could better be modeled by Lévy flights (Kuhwald and Pavlyukevich, 2016).

60    Paleoclimatic data indicate the coincidence of transitions from strong to weak or from weak to strong thermohaline circulation states with the occurrence of extreme climatic variations (Vélez-Belchí et. al., 2001). In our previous work (Tesfay et. al., 2020), we investigated the most probable trajectories of such transitions. The results of this study led us to a number of questions, of how is the overturning circulation stable to abrupt climatic changes? And what parameters of Lévy noise most affect the equilibrium of the AMOC? We also estimate possible scenarios for thermohaline circulation regeneration, analyzing

65    the stability of the *off* state under stochastic fresh-water fluctuations.

We analyze the stability of metastable states of the AMOC model by calculating three quantities, namely, mean first passage (exit) time, first passage (escape) probability and stochastic basin of attraction that carry the dynamical information of the model. In the present form of the AMOC, analyzing the intensity and mechanism of the forcing schemes that could trigger such transitions and studying the stability of strong and weak AMOC equilibrium states is of fundamental importance.

70    Particularly, we will study the effect of extreme events on the scalar stochastic AMOC model

$$dY_t = -V'(Y_t)dt + dL_t^\alpha \tag{1}$$

by measuring the stability of equilibrium states of the salinity difference process $Y_t$ for various values of (nondimensional) fresh-water forcing and non-Gaussianity parameter $\alpha$. In Eq. (1), $V$ is a potential function (details are given in Section 2).

The paper is structured as follows. In Section 2, we discuss the simplified conceptual stochastic Stommel two-compartment

75    model for AMOC. A brief introduction of the stability analysis measures is provided in Section 3. Stability analysis of the stochastic overturning circulation system is given and results obtained are presented in Section 4. Our findings are summarized in Section 5.

**2   Atlantic meridional overturning circulation model**

Thermohaline circulation is an oceanographic phenomena that refers to the movement of ocean waters across both hemispheres

80    and is responsible for the heat transfer and redistribution, acting as a regulator of the global climate. The schematic functioning of AMOC is shown in Fig. 1. The main engine of this circulation is the difference in density between ocean currents $\Delta\rho$, which is determined by the salinity $S_e, S_p$ and the temperature $T_e, T_p$ of the water and can be represented by

$$\Delta\rho = \rho_0[\beta_S(S_e - S_p) - \beta_T(T_e - T_p)], \tag{2}$$

where $\beta_T = 0.17 \times 10^{-3}\ ^o\ C^{-1}$ is the thermal expansion coefficient and $\beta_S = 0.75 \times 10^{-3}$ psu$^{-1}$ is the haline contraction

85    coefficients, respectively. The surface ocean waters in the subtropical regions due to intense evaporation $F_s/2$ have high salinity $S_e$, however the high water temperature $T_e$ maintains the low density and prevent surface waters sinking.

In high latitude areas, the formation of dense water is mainly associated with lower temperatures $T_p$ and increased salinity $S_p$ due to the formation of ice. Thus, in the polar regions, the increase in surface water density causes it to sink and displace

[Figure]

**Figure 1.** The two-compartment model of Stommel adapted from (Cessi, 1994). Each compartment represents the waters of the equatorial and polar oceans with the same volumes $V_c = 300 \times 4.5 \times 8,250 \text{ km}^3$ and temperature $T_e/T_p$ and salinity $S_e/S_p$ characteristic of each of them. The other system parameters are the mean ocean depth $H = 4500$ m, the exchange mass function $Q$, the density gradient $\Delta\rho$, the freshwater flux $F_s$, the equatorial atmospheric temperature $T_{ae}$, the polar atmospheric temperature $T_{ap}$, the reference temperature $T_0 = 5°$ C, the reference salinity $S_0 = 35$ psu, the meridional temperature difference $\theta = 25$ K and the temperature relaxation time scale $t_r = 25$ days.

deep water. In this way, the origin of the thermohaline circulation is a vertical flow of surface water $\frac{1}{2}Q(\Delta\rho)$, diving to an
90   intermediate depth or close to the bottom, depending on the density of that water. The systems of superficial and deep circulation of the oceans are interconnected. The continuation is a horizontal flow: the recently sunk waters repel in the horizontal direction the deep waters that occupied this place. In this way, the cold, dense waters sink and slowly flow towards the equator. Thermal energy and salinity balances can be defined by the system of differential equations (Cessi, 1994) (the dots represent derivative with respect to time):

95
$$\dot{T}_e = -t_r^{-1}(T_e - (T_0 + \frac{\theta}{2})) - \frac{1}{2}Q(\Delta\rho)(T_e - T_p),$$
$$\dot{T}_p = -t_r^{-1}(T_p - (T_0 - \frac{\theta}{2})) - \frac{1}{2}Q(\Delta\rho)(T_p - T_e),$$
$$\dot{S}_e = \frac{F_S}{2H}S_0 - \frac{1}{2}Q(\Delta\rho)(S_e - S_p),$$
$$\dot{S}_p = -\frac{F_S}{2H}S_0 - \frac{1}{2}Q(\Delta\rho)(S_p - S_e), \tag{3}$$

where $Q(\Delta\rho) = t_d^{-1} + V_c^{-1}q(\Delta\rho)^2$ is the exchange mass function with diffusive time scale $t_d = 180$ years between the two compartments and transport coefficient $q = 1.92 \times 10^{12}$ m$^3$s$^{-1}$. The other parameters of the system are defined in the caption of Fig. 1.

The time evolution of temperature $\Delta T \equiv T_e - T_p$ and salinity difference $\Delta S \equiv S_e - S_p$ between the compartments are obtained by subtracting the conservation equations (3), respectively.

$$\frac{d\Delta T}{dt} = -t_r^{-1}(\Delta T - \theta) - Q(\Delta\rho)\Delta T,$$

$$\frac{d\Delta S}{dt} = \frac{F_S}{H}S_0 - Q(\Delta\rho)\Delta S. \tag{4}$$

The original system (4) with the substitutions $\quad x \equiv \frac{\Delta T}{\theta}, \quad y \equiv \frac{\Delta S\beta_s}{\theta\beta_T}, \quad \tau \equiv \frac{t}{t_d}$ is reduced to the dimensionless system of evolution equations (Cessi, 1994; Tesfay et. al., 2020)

$$dx = (-\beta(x-1) - x[1 + \mu^2(x-y)^2])d\tau,$$

$$dy = (F - y[1 + \mu^2(x-y)^2])d\tau, \tag{5}$$

where $\beta$ a ratio of a diffusive and a restoring time scale, $\mu^2$ strength of the buoyancy-driven convection between the two compartments relative to the diffusive mixing and $F$ dimensionless fresh-water forcing are defined as

$$\beta = \frac{t_d}{t_r}, \qquad \mu^2 = \frac{qt_d(\beta_T\theta)^2}{V}, \qquad F = \frac{\beta_S S_0 t_d}{\beta_T\theta H}F_S. \tag{6}$$

[revised manuscript text omitted]

$$\nu_\alpha(dz) = \frac{\alpha \, \Gamma((1+\alpha)/2) \, dz}{2^{1-\alpha} \sqrt{\pi} \, \Gamma(1-\alpha/2) \, |z|^{1+\alpha}}, \quad \alpha \in (0,2), \tag{11}$$

where $\Gamma$ is the Gamma function. When $\alpha \in (0,1)$, the $\alpha$-stable Lévy motion has finite variation, otherwise, when $\alpha \in [1,2)$ it is unbounded.

Comparing two stochastic processes such as Brownian motion and Lévy flight, several differences and some similarities can be distinguished. This comparison will once again justify our choice of noise for simulating climate extreme events.

Both processes are a random walk with independent and stationary increments. Brownian motion increments have a Normal distribution, while Lévy flight step-lengths have a Lévy distribution, a probability distribution that is heavy-tailed. The sample path of the Brownian motion is continuous, differing in this from Lévy flight since its path is a cadlag function. Namely, Lévy path at most has a countable number of jumps, which are the only discontinuities in time.

**3.2 Mean first exit time**

The first exit time for a solution orbit from a deterministic domain $D \subset \mathbb{R}^1$ of attraction of $y_{(on/off)}$ is defined as:

$$\tau(\omega, y) = \inf\{t \geq 0, Y_t(\omega, y) \notin D\}, \tag{12}$$

and the mean exit time or the mean residence time of the process in the *on(off)*-state domain is denoted as $u(y) \triangleq \mathbb{E}\tau(\omega, y) \geq 0$. It has been proven (Duan, 2015) that the mean exit time of the stochastic system (9) for an orbit starting at $y \in D$, satisfies the following nonlocal partial differential equation with an external boundary condition

$$Au(y) = -1, \quad y \in D$$
$$u(y) = 0, \quad y \in D^c, \tag{13}$$

where $A$ is the generator defined as

$$Au(y) = -V'(y)u'(y) + \int_{\mathbb{R}^1\setminus\{0\}} [u(y+z) - u(y) - I_{\{|z|<1\}} \, zu'(y)]\nu_\alpha(dz). \tag{14}$$

Here $D^c$ is the complement set of $D$ in $\mathbb{R}^1$ and $I_{\{|z|<1\}}(z)$ is the indicator function for a set $|z| < 1$, defined as

$$I_{\{|z|<1\}}(z) = \begin{cases} 1, & \text{if } |z| < 1, \\ 0, & \text{if } |z| \geq 1. \end{cases} \tag{15}$$

Moreover, the generator can be interpreted as $Au = \lim\limits_{t \to 0} \frac{\mathbb{E}u(y_t) - u(y_0)}{t}$, for every $u \in C^2(\mathbb{R}^1)$.

**3.3 Escape probability**

The likelihood that the salinity difference process $Y_t$ exits firstly from the *on(off)*-state domain $D$ by landing in the set $U \in D^c$ belonging to the *off(on)*-state domain is represented by

$$p(y) = \mathbb{P}\{Y_\tau(y) \in U\} \tag{16}$$

and solves the following integro-differential equation with the Balayage-Dirichlet boundary condition

$$Ap(y) = 0, \quad y \in D, \tag{17}$$

$$p(y) = \begin{cases} 1, & y \in U, \\ 0, & y \in D^c \setminus U. \end{cases}$$

We use a numerical approach adapted from Gao et. al. (2014) for solving equation (17). The simplified form of this equation

$$\frac{d}{2}p''(x) + f(x)p'(x) - \frac{\varepsilon C_\alpha}{\alpha}\left[\frac{1}{(1+x)^\alpha} + \frac{1}{(1-x)^\alpha}\right]p(x)$$

$$+\varepsilon C_\alpha \int_{-1-x}^{1-x} \frac{p(x+y) - p(x)}{|y|^{1+\alpha}} dy = 0, \tag{18}$$

is discretized in the $x \in (-1,1)$. $p(x) = 1$ for $x \in (1, +\infty)$ and $p(x) = 0$ for $x \in (-\infty, -1)$. The corresponding schemes for the general case can be extended easily, using central difference for derivatives and the "punched-hole" trapezoidal rule getting a discretized equation of second-order accuracy for any $0 < \alpha < 2$ and $j = -J+1, ..., -2, -1, 0, 1, 2, ..., J-1$:

$$C_h \frac{P_{j-1} - 2P_j + P_{j+1}}{h^2} + f(x_j) \frac{P_{j+1} - P_{j-1}}{2h}$$

$$-\frac{\varepsilon C_{Pj}}{\alpha} \left[ \frac{1}{(1+x_j)^\alpha} + \frac{1}{(1-x_j)^\alpha} \right] + \varepsilon C_\alpha h \sum_{k=-J-j, k \neq 0}^{J-j} {}'' \frac{P_{j+k} - P_j}{|x_k|^{1+\alpha}} = 0 \tag{19}$$

where $P_j$ is the numerical solution of $p$ at $x_j$. The interval $[-2, 2]$ is divided into $4J$ subintervals and $x_j = jh$ for $-2J \leq j \leq 2J$ integers, where $h = 1/J$. The modified summation symbol $\sum''$ means that the quantities corresponding to the two end summation indices are multiplied by $1/2$. For more information see Gao et. al. (2014).

**3.4  Fokker-Planck equation**

[revised manuscript text omitted]

Lévy motion with small jumps and high frequency ($\alpha = 1.5$) contributes to faster escape compared to the events characterized by moderate or big jumps that correspond to moderate or low probability ($\alpha = 1$ or $0.5$). Actually, when rate of fresh-water released is large (Jackson and Wood, 2017) the timescale of AMOC weakening is measured in decades and does not depend on the actual rate of fresh-water inflow, because advective feedbacks become enhanced.

For strong fresh-water forcing the potential function $V(y)$ is also asymmetric, but now with the reversed depth: *off*-state $y = 1.14$ here is deeper than *on*-state $y = 0.37$. In this model Fig. 3 (b), *off*-state is more stable than *on*-state and this implies that under the influence of stochastic noise, the transitions from *on* to *off* are more expected.

Comparing the dimension of the effects that noise with different $\alpha$ parameter values causes in the meridional overturning circulation, it should be noted that the extreme events modeled by Lévy motion with $\alpha = 0.5$ does not reduce the stability of the *off*-state significantly, inducing only $0.114$ unit cutback in the basin width. The noise with the short jumps and the high frequency destabilizes the *off*-state more than the extreme events with moderate jumps and frequency. The long-term stability (450-year duration of the model integration) of AMOC *off*-state was also revealed in the study of eddy-permitting climate model (Mecking et. al., 2016) and explained by the combination of the anomalous northward freshwater transport with the freshening due to reduced evaporation in this region.

[revised manuscript text omitted]

**5   Conclusion**

Fluctuations in the AMOC patterns influence climate, in its turn civilization-induced and natural climatic changes have impacts on AMOC. The meridional overturning circulation is bistable only for some domain of the nondimensional fresh-water

285   parameter values. Extreme events such as greenhouse gas emissions, collapse of major ice sheets and global warming induce a transition between both states. It has, therefore, great importance to sufficiently scrutinize the stability of strong (weak) AMOC states against these extreme events. In this work, we have performed analysis of stability of the *on*-state and *off*-state in the stochastic two-compartment Atlantic meridional overturning circulation model by considering three different values of the fresh-water input control parameter.

[revised manuscript text omitted]

Marotzke, J.: Abrupt climate change and the thermohaline circulation: mechanisms and predictability, Proceedings of the National Academy

360   of Sciences of the USA, 97, 1347-1350, https://doi.org/10.1073/pnas.97.4.1347, 2000.

Mecking, J., Drijfhout, S., Jackson, L. and Graham, T.: Stable AMOC off state in an eddy-permitting coupled climate model, Clim. Dyn., 47, 2455-2470, https://doi:10.1007/s00382-016-2975-0, 2016.

Mikolajewicz, U. and Maier-Reimer, E.: Internal secular variability in an ocean general circulation model, Clim. Dyn., 4, 145-156, https://doi.org/10.1007/BF00209518, 1990.

365   Rahmstorf, H.: Thermohaline circulation: the current climate, Nature, 421(6924):699, https://doi.org/10.1038/421699a, 2003.

Rahmstorf, S.: Multiple convection patterns and thermohaline flow in an idealized OGCM, J. Clim., 8, 3028-3039, https://doi.org/10.1175/1520-0442(1995)008<3028:MCPATF>2.0.CO;2, 1995.

Rahmstorf, S.: On the freshwater forcing and transport of the atlantic thermohaline circulation, Clim. Dyn., 12, 799-811, https://doi.org/10.1007/s003820050144, 1996.

370   Serdukova, L., Zheng, Y., Duan, J., and Kurths, J.: Stochastic basins of attraction for metastable states, Chaos, 26, 073117, https://doi.org/10.1063/1.4959146, 2016.

Sgubin, G., Swingedouw, D., Drijfhout, S., Mary, Y. and Bennabi, A.: Abrupt cooling over the North Atlantic in modern climate models, Nature Communications, 8, 14375, https://doi:10.1038/ncomms14375, 2017.

Song, Y., Zhang, X.: Regularity of density for SDEs driven by degenerate Lévy noises, Electronic Journal of Probability, 20,

375   https://doi.org/10.1214/EJP.v20-3287, 2015.

Stommel, H.: Thermohaline convection with two stable regimes of flow, Tellus, 13, 224-230, https://doi.org/10.1111/j.2153-3490.1961.tb00079.x, 1961.

Tesfay, D., Wei, P., Zheng, Y., Duan, J., and Kurths, J.: Transitions between metastable states in a simplified model for the thermohaline circulation under random fluctuations, Appl. Math. Comput., 369, https://doi.org/10.1016/j.amc.2019.124868, 2020.

380   Timmermann, A., Gildor, H., Schulz, M., and Tziperman, E.: Coherent resonant millennial-scale climate oscillations triggered by massive meltwater pulses, J. Clim., 21, 2569-2585, https://doi.org/10.1175/1520-0442(2003)016<2569:CRMCOT>2.0.CO;2, 2003.

Trenberth, K.E. and Solomon, A.: The global heat balance: heat transports in the atmosphere and ocean, Clim. Dyn., 10, 107-134, https://doi.org/10.1007/BF00210625, 1994.

Vélez-Belchí, P., Alvarez, A., Colet, P., and Tintoré, J.: Stochastic resonance in the thermohaline circulation, Geophys. Res. Lett., 28, 2053-

385   2056, https://doi.org/10.1029/2000GL012091, 2001.

Zhang, X.: Densities for SDEs driven by degenerate $\alpha$-stable processes, Ann. Probab., 42, 1885-1910, https://doi.org/10.1214/13-AOP900, 2014.

Zheng, Y., Serdukova, L., Duan, J. et al.: Transitions in a genetic transcriptional regulatory system under Lévy motion. Sci Rep 6, 29274, https://doi.org/10.1038/srep29274, 2016.

---

## Author Comment (AC5) · 10 Jan 2021

The comment was uploaded in the form of a supplement:
https://npg.copernicus.org/preprints/npg-2020-31/npg-2020-31-AC5-supplement.pdf